# Challenging the Hero Narrative: Moving towards Reparational Citizenship Education

**Gary M. Walsh** 

School of Education, University of Glasgow, Glasgow G3 6NH, UK; g.walsh.1@research.gla.ac.uk

**Abstract:** In his book, *No More Heroes: Grassroots Challenges to the Savior Mentality*, Jordan Flaherty claims the saviour mentality exists when "you want to help others but are not open to guidance from those you want to help". According to Flaherty, the adoption of this mentality results in charitable activities at individual and community levels without broader systemic change, leaving unjust power relations unaddressed. He argues that this mentality is underpinned by racism, colonialism and capitalism, as well as an unethical and historically problematic understanding of charity. With reference to the ongoing partnership work between Scotland and Malawi, this article summarises a conceptual investigation into the possibility that Global Citizenship Education perpetuates the hero narrative. Historical, political and educational research is connected to Bhabha's theory of cultural hybridity to conceptualise a theory of Reparational Citizenship Education, in contrast to the "reciprocal" approach favoured by policy makers and charitable organisations in Scotland. It is argued that this conceptual shift involves taking the hero narrative to task and that this approach has theoretical and practical implications for the future of Global Citizenship Education.

**Keywords:** global citizenship education; white saviour industrial complex; hero narrative; cultural hybridity; reparational citizenship education; social justice

---

## 1. Introduction

The phrase "White-Saviour Industrial Complex" was coined by Nigerian-American author Teju Cole in 2012 [1]. In his critique of American foreign policy, the Kony2012 video and the Invisible Children project, Cole adapted the use of the "industrial complex", a phrase originating from President Dwight Eisenhower's farewell address from January 1961. In this address, Eisenhower warned Americans of what he saw as the increasingly influential role of the military establishment and arms industry since the second World War, or as he put it, "the acquisition of unwarranted influence, whether sought or unsought, by the military industrial complex" [2] (p. 46). The phrase gained popularity and has since been adapted for other uses and contexts (e.g., prison industrial complex). Political journalist, producer and author Jordan Flaherty builds on Cole's critique in his book, *No More Heroes: Grassroots Challenges to the Savior Mentality* [3]. In this paper, following Flaherty, I use the term *hero narrative* to represent what I see as a discursive consequence arising from saviourism. While I am sympathetic to Flaherty's argument, I am less concerned with the adoption of the saviour mentality on an individual level: instead I wish to construct an understanding of how the attitude of saviourism affects our collective understanding of past events and how this understanding is employed in education.

The purpose of the paper is to examine Global Citizenship Education (GCE), from the point of view of the hero narrative. Does GCE perpetuate the hero narrative? Can GCE be considered as a form of postcolonial pedagogy that perpetuates ideas of Western domination? Does GCE lend itself to charitable activities that focus on "feel-good" individual deeds, while simultaneously obscuring the need for radical structural change? In order to address these questions, I engage in a conceptual analysis of the hero narrative, informed by Bhabha's theory of "cultural hybridity" [4]. I locate this conceptual

analysis in a practical setting by building on previous scholarly work examining Scotland's relationship with Malawi and the development and educational work arising from the 2005 Cooperation Agreement between the two countries. I find that a misleading hero narrative focussing on the role of David Livingstone is evident in this work, and that a critical re-telling is needed. I argue that this re-telling makes the case for a *reparational* approach to citizenship education, which involves taking the hero narrative to task.

The paper is structured as follows. Section 2 briefly summarises Flaherty's book, *No More Heroes*, as an introduction to the hero narrative. Section 3 engages with critical analysis of GCE from the perspective of the hero narrative, focussing on the modern-day relationship between Scotland and Malawi to illustrate the basis of this critique. Section 4 details the historical context between Scotland and Malawi, focussing on accounts of the relationship between David Livingstone and his servants, the Malawian Magololo porters. Bhabha's theory of cultural hybridity is used to reveal the interdependencies and contingencies in this relationship that are concealed by the subsequent hero narrative. I argue that this narrative has been constructed around David Livingstone and is perpetuated in current policy and educational approaches promoted by the Scottish Government and the Scotland–Malawi Partnership. The concluding discussion in Section 5 argues that the Scotland–Malawi example shows why a reparational approach to citizenship education is required, and how such an approach could dismantle and repair the damage caused by the hero narrative.

## 2. No More Heroes: A Brief Summary

> The prototypical savior is a person who has been raised in privelege and taught implicitly or explicitly (or both) that they possess the answers and skills needed to rescue others, no matter the situation ... The savior mentality means that you want to help others but are not open to guidance from those you want to help. Saviors fundamentally believe they are better than the people they are rescuing. Saviors want to support the struggle of communities that are not their own, but they believe they must remain in charge. The savior always wants to lead, never to follow. When the people they have chosen to rescue tell them they are not helping, they think those people are mistaken. It is almost taken as evidence that they need more help. [3] (pp. 17–18)

Writing from a US perspective, the political journalist Jordan Flaherty provides an account of the saviour "mentality" as part of a rousing case for grassroots political activism. His main concern is that activists, even when well-intentioned, can do harm and perpetuate forms of injustice if their actions are misdirected, consciously or unconsciously, by saviourism. He argues that this mentality is pervasive in many settings including charities, volunteering, humanitarian efforts, international aid, the education system, journalism, popular culture and government foreign policies. Saviourism can be avoided when activists decentre themselves and their own feelings; confront their own privilege and the benefits they have accrued from colonialism; do not seek to lead but to listen and follow; stand in solidarity with those affected by injustice who are already resisting; and develop an intersectional understanding of systemic oppression based on race, class, gender and ability.

Flaherty's vision is one of radical social change achieved by means of "accountable activism". Informed by postcolonial literature and the theories of Edward Said and others, he begins with an account of US colonial dominance, developing this into a critique of neoliberal capitalism, white supremacy and systemic racism. He charts a history of white saviourism, emanating from the Crusades and the colonisation of the Americas. He describes the violence and genocide of these imperialist endeavours, noting that they were driven by self-interest, while the accompanying narrative was one of saving the barbaric and uncivilised natives. Modern-day US imperialism, according to Flaherty, takes the form of charitable activities, humanitarian interventions and international aid programs that perpetuate unjust hierarchies or create financial dependencies and exploitative economic relationships. As Flaherty puts it:

If we are not challenging our colonial relationships to the so-called developing world, all our charitable efforts just make for a kinder colonialism. [3] (p. 51)

Flaherty describes a range of rhetorical devices and strategies used to promote and maintain saviourism. One of these strategies is to conceal the cruelty of neoliberal self-reliance by framing the interventions described above, paradoxically, as charitable or kind. For example: referring to emergency food programs within the borders of the US itself, he observes that basic needs are reframed as "a gift instead of a right" (p. 24), resulting in the normalisation of destitution. Citing the aftermath to Hurricane Katrina, he describes President George W. Bush visiting New Orleans, rolling up his sleeves to help rebuild houses as part of what Flaherty sees as a public relations exercise. He claims that government obligations towards its own citizens were thus reframed as individual burdens: it was left to the kindness of communities and volunteers to rebuild after the devastation, without the full support of government resources.

Flaherty contrasts "feel-good" charitable and humanitarian activities with genuine civic engagement that seeks to enable political and social change. He sees this "feel-good" approach in the work of various organisations including the Peace Corps and international NGOs, often funded by philanthropic foundations with vested interests. He cites the "awareness-raising" initiatives of non-profit organisations such as the Save Darfur Coalition and Invisible Children; organisations that focussed respectively on human rights abuses in western Sudan and war crimes in Uganda. The activities of these campaigns, according to Flaherty, were a form of simplistic advertising using promotional films, celebrities and petitions. Citing various sources, he identifies hidden motivations of these activities, such as the obfuscation of a US anti-Palestinian and anti-Muslim political agenda, and the use of humanitarian aid to reduce the costs of war while funding military "peace-keeping" activities in Africa. He concludes that " ... these campaigns do not lend themselves to long-term engagement [because] they are all about the quick fix ... They also reinforce old stereotypes about western superiority and about Africa's need for external salvation, by calling white rescuers" [3] (p. 46).

The central thrust of Flaherty's argument is that the saviour mentality, which preserves ideas of Western dominance and superiority, characterises these efforts. While Flaherty claims that saviourism can manifest itself at all levels of social action, ranging from individual or collective acts of kindness, civil society initiatives and state policy-level responses, I find that he is less clear on precisely how this mentality traverses these levels. In this paper I suggest that saviourism is not only a mentality but that it also forms a wider discourse, the hero narrative, that influences social action. The problem as I see it is not in our heads but in the stories that we tell ourselves. A consequence of this, as discussed in the next section, is that the hero narrative has the capacity to survive even when the saviour mentality is not necessarily present.

Flaherty concludes by directing the reader towards critical activism with the goal of systemic change, led by democratic social justice movements that are attuned and accountable to the desires of those people who are most affected by injustice and who are, by definition, those who are in the best position to take informed action. Flaherty's book is a readable and convincing summary of the modern-day effects of historically located colonialism, and an effective critique of US foreign policy, humanitarian aid and charitable activities. As such, it provides a neat stimulus for educators interested in the themes of decolonisation, social justice and citizenship education. He constructs a compelling argument in favour of refocussing the narrative from "heroes" towards the perspectives of those people most affected by injustice. Beyond that, it provides a basis on which to critique the political motivations informing the creation and maintenance of the hero narrative in education policy. Applying these principles of analysis, I now consider the implications of these concepts for GCE.

## 3. Global Citizenship Education and the Hero Narrative: Political and Historical Contexts

Saviourism and the hero narrative have important implications for the analysis of GCE, which I argue should be prioritised, especially as GCE has an important role in influencing educational policy programs around the world [5]. Recognising that GCE is a contested term with no agreed definition, my

understanding is informed firstly by the general description promoted by UNESCO [6]. This holds that GCE is a "framing paradigm" that "promotes an ethos of curiosity, solidarity and shared responsibility" (p. 15), with the aim of "securing a world which is more just, peaceful, tolerant, inclusive, secure and sustainable" (p. 9). In this paper I seek to clarify what I term as a *critical historical consciousness* within this understanding of GCE, which I argue can be successfully achieved by giving consideration to the practice of reparations. Andreotti's discussion of "critical global citizenship education" is useful to my purposes here [7], as is existing literature in educational research addressing the historical contexts of social change [8,9], historical narratives and culture [10,11]. Ultimately, this is a hopeful iteration of citizenship education that is rooted in a sense of history as opportunism and not determinism [12].

Some studies have considered the role of saviourism in education, although the rubric of saviourism is not consistently used. Where it does appear, it tends to follow the theoretical lineage of Critical Race Theory (CRT) and Critical Whiteness Studies. While these theories originate in the US, my view is that they can be meaningfully incorporated to explain events in the UK. CRT continues to gain influence in the UK [13] and other theories such as "racial neoliberalism" have been used to compare race relations in the US and UK [14,15]. I am mindful that the introduction of CRT as an analytical tool to examine UK education has been critiqued on the basis that it represents an infiltration of American-left political analysis and that it risks prioritising the issue of race over socio-economic class [16,17]; however I do not engage in this debate here. I maintain that a discussion of saviourism is relevant to the UK educational context, informed in part by the shared history between the US and UK of colonial dominance, and the continuation of these endeavours through the "special relationship" and the "War on Terror" [15].

I see the hero narrative and saviourism as part of the more general "discursive turn" in the social sciences where language is used to help construct meaning around social phenomena, which also explains the shift towards postcolonialism in GCE [18]. The discourse of saviourism is one that grapples with complex histories and social contexts. It explains events and tendencies insofar as we are engaged in a negotiation of the construction of meanings, and naming the concept of saviourism means "we are choosing one possible story rather than a "universal truth" [18] (p. 236). The story that emerges from saviourism, as I refer to it here, is the hero narrative.

Straubhaar [19] identifies saviourism as an important consideration for adult educators working in the Global South. He argues that developing a "critical consciousness" is an essential task for reflective educators who wish to challenge their own preconceptions and the privileges associated with being racialised-as-white. Aronson [20] adopts a similar theoretical perspective in her work as a teacher educator. Both authors are explicit that they themselves, as "white privileged" educators, have had to work hard to overcome ingrained saviourism, and that this demands ongoing efforts involving reflection and challenging their own worldviews. Both Straubhaar and Aronson reflect on written work (field work and journal entries, respectively) from earlier in their careers, before they gained a critical awareness of these concepts, that to them demonstrates their complicity in perpetuating saviourism. This would indicate that critical reflection on saviourism and the resulting hero narrative is not only a worthwhile exercise, but that failing to do this can limit an educator's ability to engage in social justice struggles.

From a UK perspective, MacKenzie et al. [21] ask whether the practice of UK schools partnering with schools in developing countries can be described as "reciprocal partnership or politics of benevolence", a distinction originally made by Jefferess [22], which I interpret here as a discussion of saviourism. They conduct a small-scale study focussing on the views of Scottish children (age 9 to 11) involved in the Scotland–Malawi Partnership. The authors find that the pupils involved had an awareness of their own privileged circumstances in comparison to children in the impoverished country of Malawi; they were able to engage critically with issues of poverty and inequality; they articulated the need for caring dispositions; and they justified their desires to help and to be charitable from a principled position of fairness. These views reflected what the authors considered an age-appropriate level of knowledge and understanding. The authors conclude that "the pupils' references to charity

did not resort to describing their Malawian counterparts with pity or contempt, as helpless victims waiting to be rescued by Scottish donations. We saw no grounds to conclude that money rather than education is the primary goal of the activities in question" [21] (p. 135).

I build on this analysis here by foregrounding the political and historical context of this educational intervention. The Scotland–Malawi Partnership (hereafter SMP) is particularly interesting, given Malawi's history of British colonial rule and the specific link to Scotland via David Livingstone The SMP website [23] describes the relationship between Scotland and Malawi as a "156 year old friendship ... built on mutually beneficial community-to-community, family-to-family and people-to-people links. It is a relationship underpinned by mutual understanding and respect" and that the "friendship between Scots and Malawians began in 1859, with the warm welcome extended to David Livingstone and his companions when he entered what is now Malawi for the first time". However, this description of friendship ignores the colonial history between Scotland, Britain and Malawi, as well as the complex and sometimes brutal relationship between Livingstone himself and his servants, known as the Magololo, who rebelled against him in 1861 [24]. I return to the story of the Magololo later in the article.

While MacKenzie et al. [21] do not find evidence of a "politics of benevolence" among the Scottish children involved in the SMP initiative, it should be noted that it is not the purpose of their paper or indeed the SMP itself to fully address the historical or political context of the relationship between Malawi, Scotland and the British Empire. This context is addressed by MacKenzie's co-authors, Enslin and Hedge, in prior work examining the framing of equality, reciprocity, mutual benefit and good neighbourliness in the 2005 Cooperation Agreement between Scotland and Malawi [25]. This agreement established the basis of a broad programme of activity and partnership work in the areas of health, education, trade and tourism. Enslin and Hedge note that the "complex association between Scotland and Malawi ... is imbricated in both Scotland's colonial past and its contemporary aspirations to a social democratic identity as a "responsible nation" ... there is more to their history than friendship and collaboration and it also defies simplistic treatment as merely a history of colonialism, exploitation and oppression" [25] (p. 97). Enslin and Hedge are critical but sympathetic towards the ambition of Scotland being (or perhaps being seen as) a "good global neighbour". However, they highlight a "severe tension" in their review of subsequent policies on internationalisation produced by Universities Scotland that focus largely on global and economic competitiveness [25]. Enslin and Hedge interpret this as a potential threat to the principles of equality and reciprocity, and indeed to a model of citizenship education they would like to see emerging that focuses on "the importance of political, economic and cultural geography as well as the history of colonialism to foster understanding of the context of such reciprocal relationships" [25] (p. 102). It remains unclear, however, if this model of citizenship education is evident in SMP activities or indeed the policy context that prompted such activities.

My purpose here is not to question the validity of the claim that there is no "politics of benevolence" evident among the school children involved in SMP activities; nor am I claiming that the saviour mentality is generally prevalent in education. My concern is that the hero narrative is so insidious that it can survive educational activities even when such attitudes are not readily evident among participants. This concern emerges from the observation of Haste and Bermudez that "civic actors use narratives to understand their contexts and experiences (past and present), and their agency within them. Narratives carry and frame the cultural stories we draw upon to make sense, to create identity and to define boundaries and alliances" [10] (p. 428). The hero narrative manipulates our understanding of social justice and civic action by subtly distorting the frames of reference. More specifically, I suggest that the hero narrative can conceal the historical and political contexts of such actions, and that these contexts can be usefully resurfaced when the hero narrative is explicitly challenged by means of critical reflection.

Accordingly, my next move is to expand upon the political relationship between Scotland and Malawi. I draw on the political analysis of Alexander [26], who positions the Scottish government's Malawi programme as "a tool of statecraft that assists the people of Malawi, but which also forms part of the Scottish government's wider international and domestic strategy to build its reputation"

(p. 72). Alexander emphasises that while individual political actors may have compassionate goals, "in performing development assistance the primary concern of sub-state political institutions is their own power, legitimacy and authority, rather than immediate concern with those it assists" (p. 79).

Comparing the Scottish programme to other examples and theories of "sub-state public diplomacy", Alexander's analysis points to the political motivations and advantages of the Scotland–Malawi international development programme. These include i) promoting a positive image of Scotland among the international community and the Scottish electorate; ii) providing opportunities for Scottish civic society organisations, funded by the Scottish government, to work on an international basis building networks and infrastructures in other countries and global organisations such as the UN (a responsibility that would normally sit with the UK government and not the devolved Scottish government); iii) boosting the Scottish economy through the promotion of the country's investment opportunities, especially in areas of work, education, goods and services; and iv) supporting the respective ideological goals of both Scottish unionist and nationalist political parties (Alexander argues that the pro-union government led by the Scottish Labour Party was keen to demonstrate that a devolved government could have a positive international impact, while the post-2007 government led by the subsequently dominant Scottish National Party could demonstrate that a small "nation state in waiting" had the capability to increase its commitment to international development. This latter point became particularly relevant in 2011 when the UK government withdrew its Malawian foreign aid programme). It is here, I argue, that the dynamics the hero narrative become apparent. Alexander's emphases on public image, charitable activities, economic gain and political motivations are salient in Flaherty's understanding of saviourism, as described above.

The Scottish government has since renewed its commitment to Malawi with a new Partnership Agreement signed by Scottish First Minister Nicola Sturgeon and Malawian President Peter Mutharika in April 2018 [27,28]. It remains the case that the focus of the agreement and the SMP is the present-day context, where the framing of the relationship between the two countries is not historically located in colonialism but in contemporary efforts to build a reciprocal partnership between the two countries. The activities for SMP member organisations focus on issues such as providing information, supporting UK Visa applications for visitors from Milawi and supporting the growth of business, trade, investment and tourism opportunities [29]. While a detailed examination of individual SMP projects goes beyond my purposes here, this general description appears to fit with Alexander's account of the political motivations and advantages of international development work. This is not necessarily a critique of such activities—indeed it is in keeping with the stated aim of mutual benefit for Scotland and Malawi—however this demonstrates that SMP activities are about building practical alliances rather than critically examining the complicated history between Scotland and Malawi. This is an important observation in relation to the question of whether such activities perpetuate the hero narrative, which I address in Section 4.

It could be claimed, on the basis that MacKenzie et al. [21] found no evidence of saviourism among the children involved in SMP activities, that political motivations and the colonial history between Scotland and Malawi matter little in the context of GCE. The views of children participating in SMP activities are clearly important. However, the views of pupils are not the only relevant factor: it has been shown elsewhere that the interests and level of critical understanding of the teacher is a significant determinant of whether pupils are directed towards "personally responsible, participatory or justice-oriented citizenship"; the latter of which addresses the root causes and historical precedents for unequal relations of power [30]. The findings of MacKenzie et al. [21] suggest that there are gains being made in promoting personal responsibility and participatory organising among school pupils, but a critical justice-oriented perspective, with which I am primarily interested in here, remains elusive. Indeed, Enslin [31] recognises that the SMP school partnerships are not focussed on issues of global justice, framing this initiative instead as a practical and manageable response to the disparities of wealth between the two countries. However, educational engagement with people from a "poor" country is unsatisfactory if it fails to address the reasons for that poverty and the ways in which the

relative wealth of the West is implicated. In short: the absence of a politics of benevolence does not equate to the presence of critical justice-oriented consciousness.

This perspective raises important questions: can we expect schools to address global poverty? Would doing so be a case of placing a responsibility on schools that they simply cannot discharge? These questions and my considerations in this paper speak to a fundamental dilemma in GCE, described by Andreotti as "whether and how to address the economic and cultural roots of the inequalities in power and wealth/labour distribution in a global complex and uncertain system" [7] (p. 22). I recognise that engaging in hard-edged political and postcolonial critique of the relationship between Scotland and Malawi may pose a risk to the concept of reciprocal partnership favoured by policy makers and civil society organisations in Scotland. However, I take this as an opportunity to work with the idea that GCE can potentially do more than develop reciprocal relations between rich and poor countries, and that reparational relations can be fostered through a more explicit focus on social justice. Challenging the hero narrative and fostering critical awareness of the political and historical contexts of Scotland–Malawi relations, if done with care, has the potential to strengthen the bonds between the two countries.

## 4. Challenging the Hero Narrative: Confronting the Past, Retelling the Story

In this section, I return to the history of the Malawian Magololo porters, revealing the problems associated with the hero narrative of David Livingstone. This involves looking in some detail at the cultural and political history of the Magololo. I cannot offer a full account of this complex story here; instead I attempt to highlight the most relevant points. Doing so enables me to locate this story within Bhabha's theory of cultural hybridity, laying the foundation for my concluding conceptualisation of *Reparational Citizenship Education* in Section 5.

The SMP website and materials, Scottish Government documents and information from other partner organisations endorse an uncritical and heroic account of Livingstone's influence in Malawi, often citing his travels as the beginning of the relationship and shared history between the two countries. It is apparent that the purpose of mentioning Livingstone in these materials is promotional rather than educational, seeking to gain support and participation in partnership activities. For example: announcing a £6m million revamp of the David Livingstone Centre in 2017, the International Development Minister Alasdair Allan said "David Livingstone remains to this day a deeply inspirational and iconic figure to many people here in Scotland, in Africa and across the world. Principles of global humanitarianism and solidarity lay behind much of his work" [32]. Other Livingstone projects include the Livingstone Fellowship Scheme which has provided Malawian and Zambian doctors with specialist medical training in Scotland [33], and the David Livingstone Birthplace Project run by the David Livingstone Trust, who make the striking claim that "For many Sub-Saharan Africans the experience of coming to the birthplace of David Livingstone is one of spiritual pilgrimage" [34].

This would no doubt concern Flaherty [3], who suggests that the hero narrative focuses on the stories of "great men" and the "white-washing" of history, which is often unconscious and evident in what is not given due consideration in education. Bhabha identifies that heroes and settlers would not be framed in such terms without indigenous people standing in the story, "unsatisfied" [4] (p. xxi). The Magololo are concealed within their own story while David Livingstone, by contrast, is heralded as a key figure in this historical narrative. John M. MacKenzie observes that the "hero worship of David Livingstone and the countless biographies written in his memory" indicates the influence of missionaries in advancing a distinctively Scottish identity both at home and abroad [35] (p. 224).

Mandala's account seeks to recentre the actions of the Magololo in the formation of this relationship, and in the story of Africa's encounters with the British Empire and capitalist labour relations [24]. He reveals the nature of the relationship between Livingstone and the Magololo by drawing on documentary evidence and oral sources. On a theoretical level, this resonates with Bhabha's approach, which details the interdependencies and contingencies of colonising and colonised cultural identities.

Bhabha contends that these identities are developed in the "interstices", the in-between spaces, where relationships and meanings are negotiated:

> What is theoretically innovative, and politically crucial, is the need to think beyond narratives of originary and initial subjectivities and to focus on those moments or processes that are produced in the articulation of cultural differences. These "in-between" spaces provide the terrain for elaborating strategies of selfhood—singular or communal—that initiate new signs of identity, and innovative sites of collaboration, and contestation, in the act of defining the idea of society itself. [4] (p. 2)

Bhabha commends us to consider instead "the possibility of a cultural hybridity that entertains difference without an assumed or imposed hierarchy" [4] (p. 5). This presents the challenge of considering the relationship between Magololo and Livingstone with what Rancière described as a fundamental "assumption of equality" in the intelligence of all human beings [36]. By contrast, the relationship between Livingstone and the Magololo was predicated on an assumption of inequality, where Livingstone's "civilising mission" [37] and indeed that of the British Empire itself "sees emancipation as the act through which someone is made equal through an intervention from the outside" [36] (p. 51); an attitude that is closely related to saviourism. The relationship between Livingstone and the Magololo was clearly not one of fundamental equality but a case of "racialised unequal exchange" [38], as the Magololo were exploited as a form of capital and labour. This attitude of superiority is, somewhat ironically, responsible in part for the creation of Malawi as an independent state in 1964. The collapse of the Central African Federation; a "partnership hoax" created by the British government to preserve white domination in Africa, in defiance of African wishes and especially those of Nyasaland, gave rise to the independence of Nyasaland from Britain and the creation of Malawi [39].

It would not be correct, however, to claim that the actions of the British or Livingstone himself created the social inequality in Malawi. Mandala recounts the precolonial conquests of the ruling African aristocrats (the Kololo) and their king, Sekeletu, who struck a bargain with Livingstone. This bargain was strategically designed for two reasons: firstly, to enable Sekeletu to occupy the Bakota Highlands owned by the Ndebele, so that he could relocate his people there to avert an outbreak of malaria that threatened his rule. The Ndebele king had a connection with Livingstone's wife, Mary, as her father, Reverend Moffat, was a trusted friend [24]. Secondly, Livingstone had unveiled his plan to "civilise" the Kololo kingdom by introducing Christianity, as well as capitalist relations with Britain and European traders, which was dependent on opening a trade route across Africa [37]. Sekeletu enthusiastically agreed to this plan, and Livingstone's request to sponsor his journey, despite fierce opposition from his own chiefs (*indunas*), who by this time had become suspicious of Livingstone's motives. Part of Sekeletu's sponsorship deal was to provide the "unfree labour" of the Magololo porters [37]. It is apparent, then, that Livingstone's arrival was capitalised upon by Sekeletu to resolve pre-existing political tensions and inequalities. The negotiations between Sekeletu and Livingstone therefore reveal some of the interdependencies and contingencies at play, in the interstices, from the outset.

Another interdependency emerges as Mandala notes that "Livingstone could not have accomplished his mission without the Magololo, who created the possibility for his life to be memorable" [24] (p. 16). While the hero narrative and claims of friendship made by the SMP and the Scottish Government are clearly exaggerated, they are not without basis. Livingstone initially struck up a friendly relationship with the Magololo and treated them well, particularly while he was still dependent on Sekeletu's support. Additionally, he received the welcome of a "national hero" in England in 1856 after his first expedition; hence the hero narrative that lives on in British cultural memory [40]. Livingstone's second expedition along the Zambezi received financial backing from the British Government and industrial capitalists who were attracted by Livingstone's promises of opportunities in Africa, such as extracting cotton, establishing export/import routes and spreading the Christian faith. The Magololo, despite waiting for Livingstone's return, were initially excluded as this

expedition used steam technology and their skills as canoemen, elephant hunters, gold diggers and porters [41] were surplus to requirements. However, Livingstone later re-engaged the services of the Magololo as paid labourers. According to Mandala's reading, it was this transition from unfree labour (under Sekeletu, who had since died) to wage labour that forged a more instrumental relationship with Livingstone while marking a new era for the Magololo. They later used the gains of their labour (payment in the form of commodities, guns, travel and knowledge of a new system of labour) to exploit the labour of others in the region.

The Magololo freed themselves from the services of Livingstone, who had exposed them to brutal working conditions and neglected to meet their basic needs including food, by disobeying him. This happened when one of the Magololo porters, Chiputula, shot an elephant on a Sunday despite Livingstone regarding this as a "day of prayer". Mandala, cited by Jawali, collected an oral testimony that documents the exchange between Livingstone and Chiputula:

> He (Livingstone) asked: "Where have you been? A gun has just fired". Then he answered: "Yes, I have been to the forest and killed an elephant". Then he said: "Truly, indeed, you have been hunting on a Sunday like today, a day of prayer? Oh, no, you will be left here, all of you". He gave them guns and gun-powder, all of them, and that was how he left them there. And he told them: "if you should wish to return to your homeland, the way is open. Take it". [42] (p. 68)

The Magololo were now seen as troublemakers and were left at the Zambezi while Livingstone continued his ill-fated journey that ended in his failure to establish a British colony. Recounting the day that Livingstone left the Magololo, Mandala notes, "the events of that day became the centerpiece of the founding charter of the Magololo chiefdoms in Malawi" [24] (p. 31). The Magololo subsequently had cordial relations with the British, with the notable exception of conflict with George Fenwick, a former Church of Scotland missionary and employee of the African Lakes Company (ALC). Fenwick murdered the aforementioned Chiputula, who had become a Magololo chief, as part of an ALC plot to achieve economic dominance in the region by gaining control of the Elephant Marsh Game Reserve, and the lucrative elephant tusk ivory, from the Magololo [41,42].

Mandala's work on the history of wage labour in Southern Africa revolves around Livingstone but surfaces the role of the Magololo. On one level, Mandala's work is a history of the introduction of capitalist labour relations in South Africa. On another, using Bhabha's words, it is a history of "exploitation and the evolution of strategies of resistance" [4] (p. 9), as illustrated here:

> The Magololo come into view as men who had their own ambitions and dreams and as historical agents who, in pursuit of their interests, sometimes set courses of action counter to their master's plans ... [They] spent most of their time in service work whose long-term consequence would be large colonial and capitalist profits but whose short-term result was fame for Livingstone. Finally, the story of the Magololo reveals conditions under which even illiterate black workers rebelled against powerful white employers. [24] (p. 16)

Mandala reveals to us how identities and societies are formed in Bhabha's "Third Space of enunciation" [4] (p. 54). Mandala shows how Livingstone and the British owe their success and cultural identity, in part, to the Magololo; and how the Magololo used this relationship and their subsequent resistance against Livingstone and the Europeans as the basis of their rule in Malawi. This account does not glorify Livingstone or romanticise the Magololo: instead it reveals the contingencies and interdependencies of their identities and histories, while exposing the extent of the unequal power relations between them. I am careful here not to suggest that the Magololo enjoyed uncontested power after Livingstone, far from it in fact. Their rule in Nyasaland was determined by the British Empire and the continuing influence of Livingstone even after his death. Livingstone's friends and students played important roles in Scottish enterprise and missionary endeavours, such as the Free Church and African Lakes Company in Nyasaland, which were founded to advance Livingstone's ideals [35].

By foregrounding the story of the Magololo, Mandala demonstrates Bhabha's observations, influenced by Frantz Fanon, that "political survivors become the best historical witnesses" [4] (p. 12). This account survives in the oral tradition of Malawi even when it is marginalised in Eurocentric versions of history. The importance for subordinated peoples of retrieving their repressed histories and the desire for recognition is a location of agency and empowerment. The dominant narrative of Livingstone as a hero, however, oppresses the story of the Magololo, and in so doing it simultaneously conceals the role that the Magololo played in the formation of British cultural identity, as well as the colonial foundations of capitalism itself. It is in this way that the hero narrative continues the silencing and othering of the colonised. It fails to enunciate the relationship between the colonisers and the colonised and is therefore one of the "restrictive notions of cultural identity with which we burden our visions of political change" [4] (p. 55). Correcting this involves confronting the past and constructing a "subversive alternative" to the hero narrative [10]. I propose that this can be achieved by embarking on a cultural and pedagogical project that I term in the final section of this paper as Reparational Citizenship Education.

## 5. Moving Towards Reparational Citizenship Education

So far in this article I have interrogated the concept of the hero narrative, focussing on accounts of the Magololo in order to highlight some of the specific pedagogical implications of historical narratives. I contend that the story of David Livingstone is a hero narrative that obscures uncomfortable but important details that impinge on social and cultural identities. Using Bhabha's theory of cultural hybridity, I have argued that the relationship between Scotland and Malawi, like that of Livingstone and the Magololo, is interdependent and contingent. Thus, the resulting cultural identities cannot be easily separated along arbitrary lines such as coloniser/colonised, settler/native or hero/victim: claims of "purity" of any cultural identity can be challenged on these grounds [4].

The implications are pedagogical insofar as education has a bearing on how we come to interpret and understand past events and processes of agency and social change [8,9], and how historical narratives and culture are used to build identities and societies [10,11]. If historical agency is understood by students as something possessed and contested by individual powerful men and social institutions, as is the case in clichéd hero narratives, this can lead to a sense of despair and apathy among students who assume that they have limited capability to influence social change [8]. Thus, a pessimistic view emerges, holding that students, teachers and education itself are at the mercy of global society, and can either reproduce the existing ideology or make "pragmatic accommodations" to it [12].

In this concluding section of the article I construct a conceptual response to the hero narrative: Reparational Citizenship Education (RCE). I do so with some trepidation as I am conscious that the field of education is beset by acronyms and jargonistic catchphrases. I would urge the reader not to view RCE as a (re)branding exercise but rather a discursive tool that enables consideration of the ways in which citizenship education can be understood to have reparational capabilities and goals. My specific purpose is to explore how, through educational endeavours, we might repair the damage caused by the hero narrative. Before addressing this, however, I explore how we might begin to dismantle it.

### 5.1. Dismantling

As discussed earlier in this article using the text of Flaherty [3], the hero narrative is employed to offset possible challenges to institutional racism and white domination. The hero narrative constructs a story that glorifies individuals and the societies they represent. Dismantling that narrative, however, does not involve merely correcting or dismissing it. To the contrary, examining the power and attraction of the hero narrative reveals the cultural and political processes that have enabled the story to take hold, and why alternatives stories are repressed. As Temin and Dahl put it:

> . . . our concern is not with getting our narratives "right" but with exploring how different historical narratives differently constrict or enable certain political approaches to historical injustice. [43] (p. 906)

Saviourism and the hero narrative centre the identities of oppressors, as part of a history of the West, without identifying or giving voice to those who are apparently being saved, as in the case of Livingstone and the Magololo. It is this historical deceit, and the narrative of glory that continues to support it, that needs to be dismantled if we are to create new relationships based on equality. The glorified narrative of patriotic history has been deliberately constructed and maintained through the promotion of national curricula, most readily observed in the UK by the education policies of the Thatcher government [35]. Moreover, the individualistic hero narrative shapes civic identities by promoting a "personification of national consensus" as the true and uncontested version of events, limiting the possibility of alternative explanations [8]. This has resulted in many concerns being raised by postcolonial scholars, as well as conflict relating to the goals of history and citizenship curricula, contesting the educational ground between developing critical and interpretive skills among young people and inculcating a sense of patriotism [44]. Thus, in post-colonising countries such as Britain, the "official" interpretation of the past presented in educational curricula has tended to reflect the desired ideological construction of the present. This includes overlooking the "dark days of the past" while seeking to celebrate and transmit the values and norms of the colonisers (ibid.). By contrast, my argument is supportive of the view that civic action should be constructed from a place of critical understanding rather than a patriotic view of the good citizen [10].

I should note at this point that Scotland's education system has been broadly separate from that of the UK since the 1707 Act of Union. However, it would be naïve to consider the historical development of the Scottish education system and curriculum in a way that is completely untethered from British history and UK policy making. Indeed, the autonomy of the Scottish education system has been "renewed or diluted" in various negotiations and conflicts with the UK Government since the Act of Union [45] (p. 139). On a broader level, while the administration and legal systems of the British Empire were "predominantly English", Scotland's role cannot be neatly separated as the Scots keenly exported their own civic culture to the colonies in the form of churches, schools, colleges and universities [35].

Revisionist attempts to shift the historical narrative from the hero narrative to a mode of contrition, from glory to apology, have resulted in politicians such as Gordon Brown claiming that Britons should celebrate the positive elements of their imperial past [44,46], thereby reifying the hero narrative all the more. This demonstrates that taking political responsibility necessarily involves more than simply allocating blame or making apologies. Shifting from glory to apology is not sufficient. Demands to "answer for" historical injustices can trigger defensiveness and a retreat to the safety of the hero narrative: What is required instead is the collective will to "respond to" injustices [43]. It is for this reason that I argue in favour of moving towards a reparational iteration of citizenship education. The task of dismantling the hero narrative is a long-term iterative process of shifting cultural identities from "glory to apology to reparation".

The educational task of dismantling the hero narrative involves confronting the past: developing historical consciousness by delving into the politics of history and memory and connecting past injustices to current injustices, as I have demonstrated above in the case of David Livingstone and the Magololo. Temin and Dahl argue that "the battle must be first waged at the level of public culture in an effort to bring citizens to avow the causal connections between past racial injustices and racial inequalities in the present. In this regard, reparations have "public-pedagogical significance in raising and reforming public historical consciousness" (McCarthy 2004, 765)" [43] (p. 906). The purpose of engendering this historical consciousness is to understand present injustices, such as the level of poverty in Malawi compared to the wealth of Scotland, and to negotiate shared narratives, meanings and identities.

Additionally, a view of agency can be developed that does not rely on the actions or dispositions of individual actors but on the contexts, needs and abilities of groups of people engaged in social struggles, conflict and change [8]. A pedagogical method that may be helpful here is Historical Perspective Taking, as described by Huijgen et al. [9], which in the case of the Scotland–Malawi example would involve students attempting to see through the eyes of the Magololo while gaining an understanding of the "social, cultural, intellectual and emotional settings" that shaped their lives and actions (p. 112). This view of agency can thus be extended to "coalitional agency", whereby participants can see themselves as "bound to one another"; building connections via "larger histories, examinations of power asymmetries, and situating current interrogations within a larger trajectory of intergenerational activism and solidarity" [47] (pp. 12–13). In this way, the hero narrative surrounding David Livingstone can be revealed, problematised, dismantled, and subsequently replaced by a strengthened solidarity among students and educators. In the next section I address the issue of how to respond to the hero narrative once it has been surfaced; how RCE can contribute to the reparation of the damage caused.

### 5.2. Repairing

My emphasis on reparational citizenship education begs the question: what is it that needs repairing, and what is the role of education in doing so? Fraser's theorisation of social justice suggests that social justice encompasses three essential dimensions: (i) redistribution—the economic dimension of social justice; (ii) recognition—the cultural dimension; and (iii) representation—the political dimension [48,49]. My main consideration in developing a theory of RCE is in relation to cultural identity; however as noted elsewhere in the article, this is intertwined with economic and political concerns. While it is arguable that economic and political reparations could be explored in RCE, the primary educational task, in my view, is to repair the cultural identities that have been damaged as a result of colonisation. More specifically, I argue that the identities of both post-colonising and post-colonised states need to be repaired in the teaching of citizenship education, recognising the interdependencies and contingencies as addressed above. To use the Scotland–Malawi example: it is not only a case of repairing the damages caused to Malawian identity, but to recognise how Scottish and British identities are dependent on it. Citizenship education in this sense is reparational as it refers, using Freire's words, to the "making and remaking of ourselves in the process of making history" [12] (p. 81).

Having established that RCE is focussed in the main on cultural identities, my next move is to show how this relates to the concept of reparations. Pedagogically, I understand this as an extension of Andreotti's critical version of GCE, which espouses that individuals should be empowered to reflect on issues of structural change, inequality, injustice, power and accountability [7]. I build on this position by recognising that critique is important, of course, but it does not necessarily get things done. I consider RCE as an actionable response to global injustice that can otherwise seem insurmountable to students and teachers, who are liable to experience feelings of guilt and helplessness when engaging critically with these difficult matters, as Andreotti recognises in her construction. A conceptual shift from global to reparational citizenship education is empowering as it implies that injustices are within our capabilities of corrective action. Injustice is not positioned as an endemic global truth; rather, it is timebound and within our collective capacities of redress.

Legal and moral cases made in support of economic reparations revolve around the "development and underdevelopment thesis", where it is claimed that the colonial West underdeveloped Africa by means of exploitation while utilising the gains to develop the West [50]. Reparations enable colonising states to take responsibility for past atrocities such as the extraction of wealth, resources and labour by providing compensation and promoting a sense of justice among those affected while also enabling both societies to reaffirm their fundamental values [51]. Reparations can be symbolic in nature rather than material or economic: taking the form of public apologies, the creation of shared narratives or public memorials for the purposes of acknowledgement and recognition (ibid.). In the US, an economic case for reparations framed as a "moral debt" was made by Ta-Nehisi Coates. His article focused in

the main on housing policy; however he also notes that there is more at stake than material concerns. Much more fundamental is the issue of identity, as "reparations would mean a revolution of the American consciousness, a reconciling of our self-image as the great democratizer with the facts of our history" [52].

Following the logic of cultural hybridity would suggest that we should not be too definitive in the separation of the economic/material and the cultural/symbolic. Returning for a moment to the hero narrative illustrates this point. In addition to exploiting the resources of other countries, the hero narrative is used to support related ideological and political goals at home. David Livingstone is not just celebrated as a hero in the UK but specifically as a working-class hero, whose life supports a sentimental "rags to riches" mythology, constructed in part to strengthen industrial endeavours in Glasgow: "an imperial city whose workforce was highly dependent upon the export opportunities of empire" [35] (p. 229). Indeed, it is has been argued that colonialism helped to strengthen unequal class structures as the advancement of capitalism allowed colonisers to exploit workers in post-colonised and post-colonising societies [53]. This demonstrates the intersections of economic, cultural and political dimensions of colonialism. Engaging in material reparations in the absence of addressing unequal cultural identities and class structures, therefore, is unlikely to succeed.

That said, it is the non-material side of reparations and the implications for cultural identity touched upon above by Coates that I am primarily concerned with in relation to RCE. Cultural efforts may help to prepare the ground for material reparations; indeed, efforts are underway to achieve such an aim. The Movement for Black Lives (M4BL), an activist coalition of groups across the US formed in 2014 to represent the interests of black communities, includes "cultural reparations" among its policy demands. This includes "reparations for the cultural and educational exploitation, erasure, and extraction of our communities in the form of mandated public school curriculums that critically examine the political, economic, and social impacts of colonialism and slavery ... " [54].

Similarly, Black History Month (BHM), which began in the US in the 1970s and has been officially recognised in the UK since 1987, has similar ambitions at the level of cultural reparations. Launching the initiative in Wales in 2013, Glenn Jordan [55] argued that Black history is a form of scholarship that seeks to transform representations of the past while engaging in a struggle to celebrate and gain recognition for people of African descent. He also cautioned that BHM is itself capable of producing a "reverse discourse"; another elitist hero narrative that praises African kings and tyrants while ignoring ordinary people (ibid.). It should be noted too that experiences of students of African and Caribbean descent during the delivery of BHM school lessons can include racial micro-aggressions, and that racial stereotypes can be reinforced when such activities are delivered in the absence of culturally competent teachers or pedagogical approaches [56]. Taking these cautions into account, I support Jordan's conclusion that Black history "matters", as it democratises history and contributes to positive identities among people of African descent, while noting that such activities cannot necessarily compensate for embedded institutional racism or white supremacy. Recognising and celebrating Black identities is important, of course, and this may go some way towards cultural reparations. However, this approach by itself does not address the wider systemic racism or the "whiteness-as-normal construction of Britain's past" [56] (p. 126).

It is for these reasons that I assert RCE should be directed at the intersections of cultural identities, taking historical and political matters into account. School pupils and teachers can and should address issues of the colonial past and the root causes of global inequalities. Doing so may strengthen endeavours to move us towards a situation where Western countries can begin the difficult task of accepting national responsibility, not just for the atrocities of the past, but also for the unjust global representations of those actions in the present [57]. It is important to note at this point that the approach being advanced here is likely to be a difficult process that may prompt conflicting and uncomfortable dialogue among participants, particularly those from colonising countries. RCE therefore requires careful consideration, planning and ethical reflection from committed educators who are ready to engage with a "pedagogy of discomfort" [58]. In the case of educational activities and partnership

work between Scotland and Malawi: the key task in challenging the hero narrative would be to ensure that any spirit of friendship, trust and neighbourliness that has been forged between the two countries not only survives but is strengthened by RCE.

## 6. Conclusions

In this article I have sought to construct a conceptual understanding of Reparational Citizenship Education, which I argue has implications in the policy and practice of citizenship education and for future research in this area. I have set out a theoretical basis for a mode of citizenship education that includes critical literacy but, crucially, extends beyond it to an explicit action-oriented pedagogical project imbued with a critical historical consciousness. My entry point into this debate is Flaherty's book, *No More Heroes: Grassroots Challenges to the Saviour Mentality*. I argue that the concept of saviourism is relevant to Global Citizenship Education, but that saviourism is best conceptualised not as a psychological mentality, but as a cultural discourse that I refer to as the hero narrative. The consequence is that the hero narrative can survive even when the saviour mentality is absent. I link this observation to Global Citizenship Education by locating my critique in the ongoing educational partnerships forming between Scotland and Malawi. Building on educational, political and historical research, I argue that a hero narrative focussed on David Livingstone has damaged the interdependent cultural identities that exist between the two countries, placing limitations on the prospects for social change. My response to this is Reparational Citizenship Education, which seeks to dismantle the hero narrative and repair the damage caused by it, presenting the possibility of a renewed sense of critical historical agency that can address global inequalities. As Freire observes: "History does not become immobilized, does not die. On the contrary, it goes on" [12] (p. 82). If this is correct, it follows that we cannot change the past, but we can make a new history.

**Funding:** This research received no external funding.

**Acknowledgments:** My thanks are due to the scholars I have cited in this text and to Ben Kisby for encouraging me to contribute to this special issue.

**Conflicts of Interest:** The author declares no conflict of interest.

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
