# Peer review of "Challenging the Hero Narrative: Moving towards Reparational Citizenship Education"

_societies, doi:10.3390/soc10020034_

Round 1
Reviewer 1 Report
This is an interesting paper that I enjoyed reading a great deal. The writing style is excellent and the content is thought-provoking and will be of interest to readers in the field. The paper could be published as it is, but I offer the following thoughts to try to push the author to make the paper stronger and to deal with some tensions/potential criticisms that some readers may raise in reading the paper.
To frame the comments I have raised, I am sympathetic to the author’s argument and I do not think that the issues I am raising are uncommon in similar literature in this field. However, I do think that advocates of the sort of line the author is taking need to deal with some tensions within the position the advocate. My thoughts are:
- There seems to be an inherent tension in the author’s use of collective vs. individual action. At times they present saviourism/uncritical GCE as being inherently individual, while at others they critique some forms of collective action. The issues here are complex, but there seem to be certain tensions that have become blurred in the paper. These are: (1) critical-solidarity based GCE / uncritical-saviour based GCE; (2) collective / individualism; (3) state action / civil society (NGOs, charities etc.) action. The key tension for this paper, I think, is the first (e.g. it doesn’t really matter whether action is collective or individual, as long as it is critical / rooted in solidarity) but this becomes blurred with the other two tensions.
- The author presumes, drawing on the work of Flaherty, that saviours are uncritical and that their activism is not rooted in solidarity. A sentence in the quote that starts section 2 states: ‘Saviours fundamentally believe they are better than the people they are rescuing’. Is there any evidence that this is actually true? Is there any evidence that it is true in educational settings? In all of the research I have conducted, formally and informally, in schools I have never found this to be the case. Teachers and students never view themselves as “better”. They actively try to mediate and deal with the tensions. The most that can be claimed is that their actions are misguided, do not go far enough, do not aim at the right targets etc., but these are lesser claims and are not necessarily associated with the idea that they view themselves as “better”.
- I am not against the idea of “reparational” citizenship education, but literature on citizenship education has been beset by endless adjectives when it is not clear that these offer much that is new/different/needed. For example, how does “reparational” CE differ from the sort of “critical (global) CE argued for by Andreotti. Don’t we just want/need a form of CE that is critical, active, compassionate and so on without a need to add adjectives?
Two other thoughts.
- Towards the end of the second page, the argument seems to contradict… “kinder colonialism” and “cruel philosophies under the guise of kindness”. These seem to be different in nature, but perhaps I have misunderstood.
- I think readers need more detail about how the hero narrative about David Livingstone is actually constituted in the SMP and Scottish Policy documents. There is brief mention of this at the bottom of page 6, but then the analysis moves straight into general discussion and the precise features in the documents are not made explicit.
Author Response
Thank you for your encouraging and insightful response to this paper. Please see below a note of the ways in I have (hopefully) addressed the points you raise. I have copied text directly from the revised paper, which is also attached for your reference. Thank you again for your extremely valuable comments.
1. Collective vs individual action
This has prompted an important clarification to the argument. Specifically, I have added the following to address this:
Lines 35-39: "While I am sympathetic to Flaherty’s argument, I am less concerned with the adoption of the saviour mentality on an individual level: instead I wish to construct an understanding of how the attitude of saviourism affects our collective understanding of past events and how this understanding is employed in education."
Line 125-132: "While Flaherty claims that saviourism can manifest itself at all levels of social action; ranging from individual or collective acts of kindness, civil society initiatives and state policy-level responses; I find that he is less clear on precisely how this mentality traverses these levels. In this paper I suggest that saviourism is not only a mentality but that it also forms a wider discourse, the hero narrative, that influences social action. The problem, as I see it, is not in our heads but in the stories we tell ourselves. A consequence of this, as discussed in the next section, is that the hero narrative has the capacity to survive even when the saviour mentality is not necessarily present."
2. saviours are uncritical and that their activism is not rooted in solidarity
This is related to point 1 where I introduce the idea that the cultural discourse of the hero narrative survives even when the saviour mentality is absent. See below, using the specific example of Scottish school children engaged in the Scotland Malawi partnership work.
Lines 250-261: "My purpose here is not to question the validity of the claim that there is no ‘politics of benevolence’ evident among the school children involved in SMP activities, nor am I claiming that the saviour mentality is generally prevalent in education. My concern is that the hero narrative is so insidious that it can survive educational activities even when such attitudes are not readily evident among participants. This concern emerges from the observation of Haste and Bermudez that ‘civic actors use narratives to understand their contexts and experiences (past and present), and their agency within them. Narratives carry and frame the cultural stories we draw upon to make sense, to create identity and to define boundaries and alliances’ [10](p.428). The hero narrative manipulates our understanding of social justice and civic action by subtly distorting the frames of reference. More specifically, I suggest that the hero narrative can conceal the historical and political contexts of such actions, and that these contexts can be usefully resurfaced when the hero narrative is explicitly challenged by means of critical reflection."
3. education has been beset by endless adjectives
See lines 518-522: "In this concluding section of the article I construct a conceptual response to the hero narrative: Reparational Citizenship Education (RCE). I do so with some trepidation as I am conscious that the field of education is beset by acronyms and jargonistic catchphrases. I would urge the reader not to view RCE as a (re)branding exercise but rather a discursive tool that enables consideration of the ways in which citizenship education can be understood to have reparational capabilities and goals."
how does “reparational” CE differ from the sort of “critical (global) CE argued for by Andreotti
See lines 618-627: "Pedagogically, I understand this as an extension of Andreotti’s ‘critical’ version of GCE, which espouses that individuals should be empowered to reflect on issues of structural change, inequality, injustice, power and accountability [7]. I build on this position by recognising that critique is important, of course, but it does not necessarily get things done. I consider RCE as an actionable response to global injustice that can otherwise seem insurmountable to students and teachers, who are liable to experience feelings of guilt and helplessness when engaging critically with these difficult matters, as Andreotti recognises in her construction. A conceptual shift from global to reparational citizenship education is empowering as it implies that injustices are within our capabilities of corrective action. Injustice is not positioned as an endemic global truth; rather, it is timebound and within our collective capacities of redress"
Two other thoughts:
1. “kinder colonialism” and “cruel philosophies under the guise of kindness”
I hope lines 96-103 clarifies this:
"As Flaherty puts it:
If we are not challenging our colonial relationships to the so-called developing world, all our charitable efforts just make for a kinder colonialism. [3](p.51)
Flaherty describes a range of rhetorical devices and strategies used to promote and maintain saviourism. One of these strategies is to conceal the cruelty of neoliberal self-reliance by framing the interventions described above, paradoxically, as charitable or kind. For example: referring to emergency food programs within the borders of the US itself, he observes that basic needs are reframed as ‘a gift instead of a right’"
2. more detail about how the hero narrative about David Livingstone is actually constituted in the SMP and Scottish Policy documents
See lines 343-356:
"The SMP website and materials, Scottish Government documents and information from other partner organisations endorse an uncritical and heroic account of Livingstone’s influence in Milawi, often citing his travels as the beginning of the relationship and ‘shared history’ between the two countries. It is apparent that the purpose of mentioning Livingstone in these materials is promotional rather than educational, seeking to gain support and participation in partnership activities. For example: announcing a £6m million revamp of the David Livingstone Centre in 2017, the International Development Minister Alasdair Allan said ‘David Livingstone remains to this day a deeply inspirational and iconic figure to many people here in Scotland, in Africa and across the world. Principles of global humanitarianism and solidarity lay behind much of his work.’[1] Other Livingstone projects include the Livingstone Fellowship Scheme which has provided Malawian and Zambian doctors with specialist medical training in Scotland[2], and the David Livingstone Birthplace Project run by the David Livingstone Trust, who make the striking claim that ‘For many Sub-Saharan Africans the experience of coming to the birthplace of David Livingstone is one of spiritual pilgrimage’[3].
[1] See https://www.gov.scot/news/david-livingstone-centre-revamp [accessed 27th March 2020]
[2] See https://www.gov.scot/news/david-livingstone-fellowships-expanded [accessed 27th March 2020]
[3] See https://www.david-livingstone-trust.org/david-livingstone [accessed 27th March 2020]"

Reviewer 2 Report
Overall this makes an interesting argument, building on the specific relationship between Scotland and Malawi and drawing wider lessons for education / education partnerships more generally. My suggestions are really just to encourage the author to tighten up the flow of the argument through the paper, and link to some of the more obviously relevant literature (at least the bit of it I’m aware of). In particular I think this suffers from a lack of engagement with the history education literature (although some of this is cited).
Suggestions to develop the article:
- Section 5.1 on dismantling the hero narrative makes a good point and makes reference to some useful literature but, given this section is claiming to sketch a new approach to education (RCE), it feels like it would be benefit from a slightly more detailed discussion of some aspects of the pedagogic principles developed in history education. It’s easy to gloss over the implications here in relation to how one teaches history for understanding and evaluating different perspectives / narrative constructions as a disciplinary project.
- This section also refers to Thatcher in relation to UK education, but her government was not responsible for the curriculum in Scotland.
- Similarly in section 5.2 it feels to me like the discussion needs to make a bit more of the pedagogic thinking this builds on, there is plenty of material from history and English educators, as well as multicultural curriculum design to bolster the arguments being developed.
- The final point about women seems slightly tagged on and makes it look like the author is as guilty of ignoring this area as Bhabha and Manadala. I’m not sure what to do about this unless the author can expand the point and indicate some of the authors / ideas that would bolster the kind of RCE envisaged. If not, maybe it’s just better to leave the point in a footnote. It feels odd to have as the final point that others have a blind spot that you haven’t made the space to address in your own writing.
- In section 5.2 I think there may be something to be said about ‘coalitional agency’ as developed by Bajaj (https://repository.usfca.edu/cgi/viewcontent.cgi?article=1043&context=ijhre). I’m aware of reading some material in the past that seems to suggest students in the west should feel a measure of collective guilt about the past and therefore undertake actions as though they were similar to making reparations for harm one has caused oneself. I think the debate about apologies and reparations is helped by the more positive view of forging an historically informed coalition to achieve justice. I think this complements the author’s own focus on cultural reparations and the deconstruction of uncritical relationships and identities.
- At times I find myself wishing the author was more direct in establishing their line of argument, whilst it is clear overall, there are moments when they seem to hold back, e.g. p.4 line 176 the website claim of ‘friendship’ is said to ‘risk masking’ the problematic features of the relationship… why not just say it ignores the colonial history (I’m not sure the examples of historical injustices are necessary here, and probably make more sense in section 4). This could sharpen up the argument in this section and reduce the other material that gets in the way of the flow of the argument. As a related observation, if the author’s aim is critical justice-oriented citizenship then any form of engagement with ‘poor’ people from a ‘poor’ country is unsatisfactory if it fails to address the reasons for their poverty (even if children do not show signs of saviour thinking), especially if there are sound historical and economic reasons for believing this is linked to the relative wealth of the west (years ago there was a lot of discussion about how to teach this through debates about ‘third world debt’ forgiveness, championed by development education organisations).
Author Response
Thank you for your extremely valuable comments on this paper, which I have used to revise the manuscript. The revised is attached for your reference and I have also quoted the relevant sections below.
1. more detailed discussion of some aspects of the pedagogic principles developed in history education
See lines 148-158:
"Recognising that GCE is a contested term with no agreed definition, my understanding is informed firstly by the general description promoted by UNESCO [6]. This holds that GCE is a ‘framing paradigm’ that ‘promotes an ethos of curiosity, solidarity and shared responsibility’ (p.15), with the aim of ‘securing a world which is more just, peaceful, tolerant, inclusive, secure and sustainable’ (p.9). In this paper I seek to clarify what I term as a critical historical consciousness within this understanding of GCE, which I argue can be successfully achieved by giving consideration to the practice of reparations. Andreotti’s discussion of ‘critical global citizenship education’ is useful to my purposes here [7], as is existing literature in educational research addressing the historical contexts of social change [8], [9], historical narratives and culture [10], [11]. Ultimately, this is a hopeful iteration of citizenship education that is rooted in a sense of history as opportunism and not determinism [12]."
See also lines 508-516:
"The implications are pedagogical insofar as education has a bearing on how we come to interpret and understand past events and processes of agency and social change [8], [9], and how historical narratives and culture are used to build identities and societies [10], [11]. If historical agency is understood by students as something possessed and contested by individual powerful men and social institutions, as is the case in clichéd hero narratives, this can lead to a sense of despair and apathy among students who assume that they have limited capability to influence social change [8]. Thus, a pessimistic view emerges, holding that students, teachers and education itself are at the mercy of global society, and can either reproduce the existing ideology or make ‘pragmatic accommodations’ to it [12]"
See also lines 584-595:
"Additionally, a view of agency can be developed that does not rely on the actions or dispositions of individual actors but on the contexts, needs and abilities of groups of people engaged in social struggles, conflict and change [8]. A pedagogical method that may be helpful here is Historical Perspective Taking, as described by Huijgen et al [9], which in the case of the Scotland Malawi example would involve students attempting to see through the eyes of the Magololo while gaining an understanding of the ‘social, cultural, intellectual and emotional settings’ that shaped their lives and actions (p.112). This view of agency can thus be extended to coalitional agency, whereby participants can see themselves as ‘bound to one another’; building connections via ‘larger histories, examinations of power asymmetries, and situating current interrogations within a larger trajectory of intergenerational activism and solidarity’ [39](p.12-13). In this way, the hero narrative surrounding David Livingstone can be revealed, problematised, dismantled, and subsequently replaced by a strengthened solidarity among students and educators"
I have also incorporated some of this into the conclusion (lines 691-710):
"In this article I have sought to construct a conceptual understanding of Reparational Citizenship Education, which I argue has implications in the policy and practice of citizenship education and for future research in this area. I have set out a theoretical basis for a mode of citizenship education that includes critical literacy but, crucially, extends beyond it to an explicit action-oriented pedagogical project imbued with a critical historical consciousness. My entry point into this debate is Flaherty’s book, No More Heroes: Grassroots Challenges to the Saviour Mentality. I argue that the concept of saviourism is relevant to Global Citizenship Education, but that saviourism is best conceptualised, not as a psychological mentality, but as a cultural discourse that I refer to as the hero narrative. The consequence is that the hero narrative can survive even when the saviour mentality is absent. I link this observation to Global Citizenship Education by locating my critique in the ongoing educational partnerships forming between Scotland and Malawi. Building on educational, political and historical research, I argue that a hero narrative focussed on David Livingstone has damaged the interdependent cultural identities that exist between the two countries, placing limitations on the prospects for social change. My response to this is Reparational Citizenship Education, which seeks to dismantle the hero narrative and repair the damage caused by it, presenting the possibility of a renewed sense of critical historical agency that can address global inequalities. As Freire observes: ‘History does not become immobilized, does not die. On the contrary, it goes on’ [12](p.82). If this is correct, it follows that we cannot change the past, but we can make a new history"
2. This section also refers to Thatcher in relation to UK education, but her government was not responsible for the curriculum in Scotland.
See lines 554-562 in which I explain the reason for mentioning this:
"I should note at this point that Scotland’s education system has been broadly separate from that of the UK since the 1707 Act of Union. However, it would be naïve to consider the historical development of the Scottish education system and curriculum in a way that is completely untethered from British history and UK policy making. Indeed, the autonomy of the Scottish education system has been ‘renewed and diluted’ in various negotiations and conflicts with the UK Government since the Act of Union [37]. On a broader level, while the administration and legal systems of the British Empire were ‘predominantly English’, Scotland’s role cannot be neatly separated as the Scots keenly exported their own civic culture to the colonies in the form of churches, schools, colleges and universities [28]"
3. the discussion needs to make a bit more of the pedagogic thinking
See section 1 above
4. The final point about women seems slightly tagged on
Yes, I agree. See above for the revised conclusion. I have removed the reference to gender here completely as I think this would warrant a separate paper in itself. In which case, it does seem odd to mention it here unless I intend to fully address it.
5. there may be something to be said about ‘coalitional agency’ as developed by Bajaj
Thank you for introducing me to this - I hadn't read it and yes I agree it is relevant. I note too that both Bajaj and I cite Bhabha's 'third space'.
See lines 584-595:
"Additionally, a view of agency can be developed that does not rely on the actions or dispositions of individual actors but on the contexts, needs and abilities of groups of people engaged in social struggles, conflict and change [8]. A pedagogical method that may be helpful here is Historical Perspective Taking, as described by Huijgen et al [9], which in the case of the Scotland Malawi example would involve students attempting to see through the eyes of the Magololo while gaining an understanding of the ‘social, cultural, intellectual and emotional settings’ that shaped their lives and actions (p.112). This view of agency can thus be extended to coalitional agency, whereby participants can see themselves as ‘bound to one another’; building connections via ‘larger histories, examinations of power asymmetries, and situating current interrogations within a larger trajectory of intergenerational activism and solidarity’ [39](p.12-13). In this way, the hero narrative surrounding David Livingstone can be revealed, problematised, dismantled, and subsequently replaced by a strengthened solidarity among students and educators."
6. At times I find myself wishing the author was more direct in establishing their line of argument, whilst it is clear overall, there are moments when they seem to hold back
I have addressed this generally throughout the article, for example:
Lines 212-214:
"However, this description of friendship ignores the colonial history between Scotland, Britain and Malawi, as well as the complex and sometimes brutal relationship between Livingstone himself and his servants, known as the Magololo, who rebelled against him in 1861"
As a related observation, if the author’s aim is critical justice-oriented citizenship then any form of engagement with ‘poor’ people from a ‘poor’ country is unsatisfactory if it fails to address the reasons for their poverty
See lines 311-316 (I hope you don't mind me adapting your phrasing here)
"Indeed, Enslin [27] recognises that the SMP school partnerships are not focussed on issues of global justice, framing this initiative instead as a practical and manageable response to the disparities of wealth between the two countries. However, educational engagement with people from a ‘poor’ country is unsatisfactory if it fails to address the reasons for that poverty and the ways in which the relative wealth of the West is implicated. In short: the absence of a politics of benevolence does not equate to the presence of critical justice-oriented consciousness"
I hope these revisions address your valuable comments. Thank you again for helping me to fine tune this paper, I feel that it is stronger for it.
